# Synthesis of Linseed Oil-Based Waterborne Urethane Oil Wood Coatings

**DOI:** 10.3390/polym10111235

**Published:** 2018-11-07

**Authors:** Chia-Wei Chang, Jing-Ping Chang, Kun-Tsung Lu

**Affiliations:** Department of Forestry, National Chung Hsing University, 250, Kuo-Kuang Rd., Taichung 402, Taiwan; kf05292@dragon.nchu.edu.tw (C.-W.C.); guluwalala@gmail.com (J.-P.C.)

**Keywords:** linseed oil, acetone process, waterborne urethane oil, wood coatings

## Abstract

The linseed oil glyceride (LOG) was synthesized by using a transesterification process with a glycerol/linseed oil molar ratio of 1.0. The waterborne urethane oil (WUO) wood coating was prepared by acetone process. First, dimethylolpropionic acid was reacted with hexamethylene diisocyanate (HDI) or isophorone diisocyanate (IPDI), followed by adding LOG at various NCO/OH molars of 0.7, 0.8, and 0.9, respectively, and the COOH-containing prepolymer was obtained. Then, the ionomer which was prepared by neutralizing prepolymer with trimethylamine, was dispersed by adding deionized water, and the water–acetone dispersion was obtained. Finally, the acetone was removed by vacuum distillation. In the whole synthesized process, the LOG and COOH-containing prepolymer could be steadily synthesized by FTIR analysis, and the weight-average molecular weight and polydispersity of COOH-containing prepolymer increased with an increase of NCO/OH molar ratios. During the water dispersion process of the ionomer acetone solution, the point of phase inversion was prolonged, meaning the solid content decreased with an increase of NCO/OH molar ratios. After acetone was removed, the color of WUO was milky-white, and it was weakly alkaline and possessed a pseudoplastic fluid behavior. The particle size of WUO increased with increasing of NCO/OH molar ratios, however, the storage stability was extended for HDI and shortened for IPDI synthesized with increasing of NCO/OH molar ratios.

## 1. Introduction

The application of coatings to wood surfaces protects the wood from damage, prolongs the lifecycle and carbon fixation duration of wood products, enables wood utilization to conform with principles of regeneration and sustainability, and enhances the value of the wood products by improving their surface quality and functionality. With an increase in the number of wood buildings, wood-based recreational facilities, and high-quality wood furniture in Taiwan, penetrating oil for wood finishing has received increasing attention. However, such wood coatings are formulated using unsaturated triglycerides, such as linseed oil, with the addition of metal dryers, solvents, and other additives. Therefore, the physical properties of their films are poorer than those of traditional wood coatings such as polyurethane coatings. Urethane oil (UO) coatings are formed by the introduction of urethane linkages (–NHCOO–) into the molecular chains of an unsaturated drying oil. As a result, coatings with oxidative polymerization drying characteristics and urethane bonds are obtained, which can improve the physical properties of films and reduce the curing time of coating [1,2]. However, when solvent-borne UO coatings are used, the volatile organic compounds (VOCs) present in the solvents result in environmental and ecological damage. As an increasing number of countries have established regulations on VOC emissions, the use of such solvent-borne coatings in the future will be increasingly restricted by the environmental regulations [3,4]. The development of high-quality waterborne urethane oils (WUOs) is a feasible and necessary target for commercial profits, environmental protection, and human health, as WUOs can replace traditional solvent-borne penetrating oil wood coatings, enhance the value of wood products, and ensure compliance with the environmental regulations [5,6].

With the consumption of petrochemical raw materials, the use of natural and renewable resources as raw materials combines the principles of environmental protection and sustainable development, leads to cost reductions, and promotes the creation of comfortable, safe, and healthy environments for humans [7,8,9]. In the paint and coatings industry, the use of renewable natural oils, such as tung oil, castor oil, or linseed oil, as raw materials for the syntheses of coatings, has become a trend of the future [3,7,10,11,12]. In the present study, linseed oil was used as a raw material for transesterification with glycerol (GL) to obtain linseed oil glyceride (LOG) and, subsequently, the LOG was used as polyols for respective reactions with hexamethylene diisocyanate (HDI) and isophorone diisocyanate (IPDI) with different NCO/OH mole ratios, to investigate the feasibility of the acetone processing method for the synthesis of WUO wood coatings.

## 2. Materials and Methods

### 2.1. Materials

Linseed oil (LO) was obtained from Chung-Hsing Chemical Co. Ltd. (Taichung, Taiwan) with specific gravity: 0.97 (25 °C); viscosity: 57 cP (25 °C); iodine value: 175; saponification value: 185; and acid value: 0.7 (all listed values were measured in our laboratory). Acetone, xylene, calcium oxide, and glycerol (GL) were purchased from Union Chemical Works Ltd. (Taichung, Taiwan), reagent grade. Hexamethylene diisocyanate (HDI) was purchased from ACROS Co. (Geel, Belgium), reagent grade. Isophorone diisocyanate (IPDI) and dimethylolpropionic acid (DMPA) were purchased from Merck Taiwan Ltd. (Taipei, Taiwan), reagent grade, and the DMPA was dried in an oven at 105 °C for 12 h and cooled before use. Molecular sieve, Type 4A (8–12 mesh), was purchased from J.T. Baker Co. (Radnor, PA, USA). Methanol was purchased from Aencore Chemical Co. (Surrey Hills, Australia), reagent grade. Triethylamine (TEA) was purchased from Tedia Co., Inc. (Fairfield, OH, USA), reagent grade. Dibutyltin dilaurate (DBTDL) was obtained from An Fong Development Co. Ltd. (Taichung, Taiwan), industrial grade. Deionized water was prepared in our laboratory. 

### 2.2. Preparation of Linseed Oil Glyceride (LOG)

LOG was prepared following the method reported by previous reports [13,14,15]. After mixing LO and GL with a mole ratio of 1:1, the 0.2 wt % of CaO catalyst was added, and the mixture was placed in a four-neck reaction flask with a condenser tube and thermometer for reaction in a nitrogen environment at a holding temperature of 200 °C for 3 h. Subsequently, the reaction system was rapidly cooled to 25 °C using an ice water bath to increase the monoglyceride yield, and LOG was obtained.

### 2.3. Preparation of Waterborne Urethane Oil (WUO)

HDI and IPDI were respectively mixed with DMPA at a mole ratio of 2:1 and placed in a four-neck reaction flask. The mixture was then diluted to 50 wt % with acetone, and allowed to react in a nitrogen environment at 60 °C. After the NCO content of the reaction mixture decreased to 50% of its initial NCO content, the LOG with various weights calculated based on NCO groups/total hydroxyl groups of LOG mole ratios of 0.9, 0.8, and 0.7 were respectively added, and the temperature of the reaction was maintained at 60 °C for 5 h. Subsequently, 0.08 wt % DBTDL was added, and the temperature was held at 60 °C for another 2 h. Carboxyl (COOH)-containing prepolymers were obtained as the reaction products, and TEA was added to achieve 100% neutralization to obtain ionomers containing R–COO^−^ + N–R groups. Acetone was added to adjust the solid content of the ionomer solution to 50 wt %, then the deionized water was added dropwise at a rate of 4 mL/min into the ionomer solution, which was stirred at a rate of 550 rpm. During the water dispersion process, the viscosity of the dispersion was monitored, and the point at which the mixture viscosity trend showed a sudden change, from a slow increase to an instant decrease, was identified as the phase inversion point, which is the transition point of the dispersed phase from an oil phase consisting of organic solvent to an aqueous phase. After the continued addition of deionized water to achieve the targeted solid content, acetone was removed by vacuum distillation, and the six WUOs were obtained, which were, respectively, named according to their synthesized diisocyanate type and mole ratio as HDI-0.7, HDI-0.8, HDI-0.9, IPDI-0.7, IPDI-0.8, and IPDI-0.9. The synthesis process of WUO is shown in Figure 1.

### 2.4. Measurement of LOG Properties

#### 2.4.1. Molecular Weight

The molecular weight of LOG was performed by gel permeation chromatography (GPC, Hitachi-L6200, Hitachi High-Tech Fielding Corp., Tokyo, Japan) with a Shodex KF-802.5 column (Showa Denko K.K., Tokyo, Japan) and Hitachi L-4000 UV detector (Hitachi High-Tech Fielding Corp., Tokyo, Japan) at a detection wavelength of 254 nm. Each sample was diluted to 0.1% using tetrahydrofuran (THF), and the elution was performed using THF with sample injection volumes of 20 μL and a flow rate of 1 mL/min. The polystyrene standards, with molecular weights of 168; 578; 1080; 2450; 5050; 10,100; and 22,000 g/mol, were used to establish a calibration curve for the measurement of weight-average molecular weight (*M*_w_), number-average molecular weight (*M*_n_), and polydispersity.

#### 2.4.2. Fourier-Transform Infrared Spectroscopy (FTIR) Analysis

FTIR spectra were obtained using a Perkin-Elmer Spectrum 100 spectromete (PerkinElmer, Inc., Waltham, MA, USA). Each sample was diluted to 5% (*v*/*v*) with acetone, coated on a potassium bromide (KBr) disc, dried, and scanned 16 times using transmission mode with a scan range of 4000–450 cm^−1^ and resolution of 4 cm^−1^.

#### 2.4.3. Hydroxyl Number 

The hydroxyl numbers were determined by using the acetic anhydride/pyridine method in accordance with the Standard Test Method for Hydroxyl Value of Fatty Oils and Acids (ASTM D1957).

### 2.5. Measurement of Prepolymer Properties

#### 2.5.1. FTIR Analysis 

The FTIR spectra were obtained as described in Section 2.4.2.

#### 2.5.2. Molecular Weight 

The instrument settings and measured properties were as described in Section 2.4.1. However, GPC was performed with a Shodex KF-803 column (Showa Denko K.K., Tokyo, Japan), and the calibration curve was established using polystyrene standards with molecular weights of 168; 578; 3250; 7000; 11,600; 28,500; and 66,000 g/mol.

#### 2.5.3. Solid Content

Solid contents were determined in accordance with CNS 5133 (Chinese National Standards, Taibei, Taiwan). The mass retention of 3 g sample in the 105 °C oven for 3 h was determined.

### 2.6. Measurement of Ionomer and WUO Properties

#### 2.6.1. Solid Content 

Solid contents were determined as described in Section 2.5.3.

#### 2.6.2. Viscosity 

The variation of viscosity during ionomer transition was measured with a Brookfield R/S Plus rheometer (Showa Denko K.K., Tokyo, Japan) using a C50-1 spindle (radius = 25 mm, cone angle = 1°) and a fixed shear rate of 30 s^−1^/min for 180 s. The rheological properties of the dispersions were measured using a shear rate range of 0–300 s^−1^ and a shear rate acceleration of 100 s^−1^/min.

#### 2.6.3. pH Value 

The pH values were measured using a Suntex pH meter (Sp-701, Suntex Ins. Co., New Taipei, Taiwan) under 25 °C.

#### 2.6.4. Particle Size 

Dynamic light scattering (DLS) using a Malvern Zetasizer Nano-ZS system (Malvern Ins., Worcestershire, UK) with a 4 mW helium/neon laser as the light source, a scattering angle of 173°, and a detection range of 0.66 nm–6 μm, was employed to analyze the z-average diameter and polydispersity index (PDI) of the suspension particles.

#### 2.6.5. Storage Stability 

Each WUO was placed in a capped 1 L polyethylene bottle, and stored away from sunlight at 25 °C. The liquid in each bottle was visually inspected for signs of gelation, precipitation, discoloration, or stratification.

## 3. Results and Discussion

### 3.1. Basic Properties of LOG

The main fatty acid component of linseed oil (LO) is linolenic acid (53.2%), followed by oleic acid (18.5%), while other constituents include linoleic acid (17.3%), palmitic acid (6.6%), stearic acid (4.4%), and a small amount of other fatty acids [16]. However, the linseed oil does not contain hydroxyl (OH) groups, and cannot react with diisocyanate. Therefore, transesterification of LO with glyceride (GL) was required to obtain OH-containing linseed oil glyceride (LOG). The GPC curve of LO in Figure 2 shows two signals located at molecular weights of 790 and 1664 g/mol, corresponding to the triglyceride structure in the LO, and the dimer formed from the oxidative polymerization of the LO during storage, respectively. In the LOG curve, the signals at molecular weights of 318 and 554 g/mol correspond to linseed oil monoglyceride and diglyceride, respectively. Although the signal at 790 g/mol was weaker compared with that of the LO curve, the presence of a signal indicates that a small amount of LO did not converse to glyceride through transesterification. In addition, a signal corresponding to the glyceride formed by the transesterification between the dimer and glycerol at 1189 g/mol can be observed in the LOG curve. Based on calculations, the *M*_w_, *M*_n_, and polydispersity of LOG were 871 g/mol, 620 g/mol, and 1.32, respectively, while they were 1275 g/mol, 968 g/mol, and 1.40 for LO, respectively. The results show that the *M*_w_ and *M*_n_ of LOG were both lower than of LO, which indicates that most of the LO already transesterificated to form linseed oil monoglyceride and diglyceride.

The FTIR spectra of LO and LOG are shown in Figure 3. In the LO spectrum, absorption peaks were absent within the wavenumber range of 3200–3600 cm^−1^, which indicates the absence of OH groups, while the spectrum of LOG formed after transesterification had a broad absorption peak at 3423 cm^−1^, which indicates the presence of OH groups in LOG. Absorption peaks indicating the stretching and bending vibrations of *cis* C=C–H bonds in fatty acids were observed at 3009 and 715 cm^−1^ in the spectra of LO and LOG, which shows that unsaturated fatty acid structures were still present in LOG. In the absorption spectrum of LO, absorption peaks representing C=O and C–O–C were observed at 1744 and 1165–1158 cm^−1^, respectively; however, the intensities of the corresponding peaks in the LOG spectrum were reduced. This is due to the replacement of fatty acids in linseed oil by OH groups after the transesterification reaction [13,17,18]. The hydroxyl value of LOG was determined to be 166, which is similar to the theoretical value of 177.

### 3.2. Basic Properties of Prepolymers

The FTIR spectra of COOH-containing prepolymers respectively synthesized by the reactions of different diisocyanates (HDI or IPDI), DMPA, and different weights of LOG based on different NCO/OH mole ratios, are shown in Figure 4. In the prepolymer spectra, the absorption peak at 3423 cm^−1^, which represents the stretching vibrations of OH groups, had a lower intensity compared with the corresponding peak in the LOG spectrum. In addition, an absorption peak corresponding to the stretching vibrations of N–H could be observed in the prepolymer spectra at 3350 cm^−1^, which represents the characteristic peak of N–H in the urethane linkages (–NHCOO^–^) formed from the reaction between OH groups in LOG and diisocyanate. The absorption peak at 1710–1730 cm^−1^ corresponds to the stretching vibrations of carbonyl groups (C=O), while the peaks at 1534 and 774 cm^−1^ correspond to the bending vibrations of N–H, and the peaks at 1245 and 1040 cm^−1^ correspond to the stretching and vibration peaks of C–O. It was also observed that the C=O absorption peak in the LOG spectrum at 1745 cm^−1^ shifted to a lower wavenumber range of 1740–1660 cm^−1^ in the prepolymer spectra. This is consistent with the results obtained in a study on castor-oil-based two-component waterborne polyurethane coatings [19]. Based on the study of the FTIR spectrum of DMPA, the C=O absorption peak of DMPA is located at 1691 cm^−1^, which proves that the prepolymers contained the COOH structure of DMPA [20]. These results indicate that COOH-containing prepolymers were synthesized through urethane linkages, which were formed by the reaction of LOG, DMPA, and diisocyanate. In addition, absorption peaks corresponding to the stretching and vibration of NCO groups were not observed at 2270–2280 cm^−1^ in the spectra, which shows that the NCO groups completely consumed.

The previous study reported that the free N–H and C=O absorption peaks in urethane linkages were located at 3440–3450 and 1730–1740 cm^−1^, respectively [21]. However, with the formation of hydrogen bonds between urethane units, the absorption peaks of N–H and C=O will shift towards lower wavenumbers. Therefore, the FTIR spectra of the COOH-containing prepolymers synthesized using different NCO/OH mole ratios were partially enlarged for further analysis, as shown in Figure 5. The spectra of the COOH-containing prepolymers synthesized with HDI and different NCO/OH mole ratios are shown in Figure 5A. The absorption peak at 3650–3250 cm^−1^ corresponding to the stretching vibrations of N–H remained unchanged. However, from the spectra of the COOH-containing prepolymers synthesized using IPDI and different NCO/OH mole ratios shown in Figure 5B, the maximum value of the N–H absorption peak for IPDI-0.7 was located at 3343 cm^−1^, while the maximum values for IPDI-0.8 and IPDI-0.9 were located at 3340 cm^−1^ and 3330 cm^−1^, respectively. Therefore, with an increase in the NCO/OH mole ratio, the N–H absorption peak shifted towards a lower wavenumber, which indicates an increase in hydrogen bonding between urethane linkages in the hard segments of the COOH-containing prepolymers. In Figure 5C, which shows the FTIR spectra of COOH-containing prepolymers synthesized using HDI and different NCO/OH mole ratios, the broad absorption peak at 1700 cm^−1^ corresponds to C=O affected by hydrogen bonds, while the absorption peak at 1740 cm^−1^ corresponds to the stretching vibrations of free C=O groups. It was observed that the locations of the absorption peaks for the three NCO/OH mole ratios were unchanged. However, the peak intensity at 1700–1740 cm^−1^ increased as the NCO/OH mole ratio increased, which indicates that urethane linkages within the molecular chains increased with an increase in the NCO/OH mole ratio. In the spectra of prepolymers synthesized using IPDI and different NCO/OH mole ratios in Figure 5D, the intensities of the absorption peaks at 1700 cm^−1^ increased with an increase in the NCO/OH mole ratio. In view of the results described above, the number of N–H and C=O groups affected by hydrogen bonds in the COOH-containing prepolymers synthesized using HDI were independent of the NCO/OH mole ratio. This is because HDI, which contains alkane chains, has greater compatibility with the fatty acid chains of the glyceride, resulting in a less orderly arrangement of molecules in the hard and soft segments of the polyurethane prepolymers. Therefore, the hydrogen bonds were not affected by the NCO/OH mole ratio used during synthesis. By contrast, IPDI has a lower compatibility with the fatty acid chains of glyceride due to the alicyclic structure in IPDI. Therefore, the prepolymers synthesized using IPDI had a more orderly arrangement of molecules in the hard and soft segments, i.e., the microphase-separated structure formed from the arrangement of soft and hard segments was more distinct. As a result, an increase in the NCO/OH mole ratio led to an increase in the number of N–H and C=O groups affected by hydrogen bonds in the prepolymers [22].

Results of GPC analysis of the COOH-containing prepolymers synthesized using different diisocyanates and NCO/OH mole ratios are shown in Table 1. Among the prepolymers, HDI-0.7 had the lowest *M*_w_ and *M*_n_ of 4214 and 1631 g/mol, respectively, followed by HDI-0.8 with *M*_w_ and *M*_n_ of 7181 and 1678 g/mol, while the highest *M*_w_ and *M*_n_ of 10,734 and 1682 g/mol were obtained in HDI-0.9. For the prepolymers synthesized by HDI, the *M*_w_, *M*_n_, and polydispersity increased as the NCO/OH mole ratio increased. Similar trends were observed in the molecular weights and polydispersity of prepolymers synthesized by IPDI, whereby IPDI-0.7 had the lowest *M*_w_ of 2962 g/mol, followed by IPDI-0.8 with a *M*_w_ of 5861 g/mol, and IPDI-0.9 with a *M*_w_ of 6126 g/mol. Due to the reactive difference of two NCO groups in IPDI, the IPDI-based prepolymer, which was obtained through two-step addition reaction, is more homogeneous than the HDI-based prepolymer. Therefore, the IPDI-based prepolymers had a lower *M*_w_ and polydispersity than the IPDI-based prepolymers.

### 3.3. Variations of Ionomer Properties during the Water Dispersion Process

As shown in Figure 1, ionomers were obtained by the neutralization of COOH-containing prepolymers with TEA, and dispersions were formed through the addition of deionized water. The phase inversion point during the water dispersion process is usually determined from variations of viscosity, conductivity, and torque of mixer [23]. In the present study, the viscosity variations of the dispersions were used to determine the phase inversion points of the water–acetone dispersions of the respective ionomers, and to confirm the transition of water-in-oil (W/O) dispersions to oil-in-water (O/W) dispersions. The previous studies [14,19,24] reported that solid content, neutralization degree, hydrophilic group content, molecular weight, and mixing shear force influenced the water dispersion process of waterborne polyurethane (WPU). Therefore, in the present study, the influence of different NCO/OH mole ratios on the phase transition under conditions of identical solid content, neutralization degree, and shear force were investigated, and the results are shown in Figure 6 and Figure 7. When deionized water was initially added, the associations between ionomers were broken, which led to a slight decrease in the viscosity of the dispersion. As the added amount of deionized water increased, the volume of the aqueous phase increased, causing the viscosity of the dispersion to increase. A sudden decrease in the viscosity of the dispersion, with the continuous addition of deionized water, signified the transition of the dispersed phase, from an oil phase to an aqueous phase, i.e., the W/O dispersion transformed into an O/W dispersion. At this point, the continuous phase was the aqueous phase, therefore, the dispersion had a lower viscosity [25].

The viscosity variation for the water–acetone dispersions of HDI ionomers in Figure 6 shows that the phase inversion points for HDI-0.7 and HDI-0.8 occurred within a solid content range of 37%–38%. When the solid content of the dispersion reached 32%, large variations in viscosity were no longer observed. Therefore, a solid content of 32% was considered the endpoint of the phase inversion. For HDI-0.9, phase inversion occurred later at a solid content of 32%, while viscosity no longer varied substantially at a solid content of 22%. Figure 7 shows the viscosity variations during the water dispersion process of the respective IPDI ionomers. The phase inversion points for the water–acetone dispersions of IPDI-0.7 and IPDI-0.8 occurred within a solid content range of 39%–40%, while the phase inversion point of IPDI-0.9 occurred at a solid content of 31%. Therefore, a solid content of 30% was considered the endpoint of phase inversion for IPDI-0.7 and IPDI-0.8, while the endpoint for IPDI-0.9 was 25%. These results show that an increase in the NCO/OH mole ratio during synthesis led to a delay in the occurrence of the phase inversion point. This is attributed to the increase in the molecular weight of the ionomer as the NCO/OH mole ratio increased, which induced the possibility of molecular chains entanglement, interpenetration, and aggregation. Consequently, the dispersed phase had poorer colloid compressibility and a larger volume, resulting in a higher viscosity after the water dispersion process, hindering the phase inversion process.

The viscosities of the water–acetone dispersions during the water dispersion process of HDI ionomers were higher than that of the IPDI ionomers. For example, the maximum viscosity of HDI-0.9 was 2224 cps, which was much higher than the maximum viscosity of 1106 cps of IPDI-0.9. This is because the prepolymers synthesized by HDI had higher molecular weights than those synthesized by IPDI. Similarly, the TEA-neutralized HDI ionomers also had higher molecular weights compared with the IPDI ionomers. Additionally, HDI is mainly composed of long alkane chains, while IPDI is an alicyclic compound. Therefore, under the same shear force, the molecular chains synthesized by using HDI were more flexible and easier to entangle, which led to higher viscosities. As IPDI molecules are more rigid, the synthesized molecular chains had lower activities, which led to a lower tendency of the chains to entangle, or compress to form spherical particles. Consequently, the viscosity variations were smaller, and the overall viscosity of the ionomer–acetone continuous phase during the initial water dispersion process was higher. When the phase inversion point of the water–acetone dispersion was reached, the continuous phase changed to an aqueous phase. However, as the water molecules of the continuous phase mostly encased the dispersed phase in the form of bound water, the suspension particles of the dispersed phase still easily entangled, which resulted in a slower reduction in the viscosity of the water–acetone dispersion of HDI-0.9 compared with IPDI-0.9. The six ionomers exhibited a clear yellow color. However, as the added amount of deionized water increased, the water–acetone dispersions became cloudy and milky. After the water dispersion process was complete, the acetone was removed from the water–acetone dispersion by vacuum distillation to obtain the respective waterborne urethane oils (WUOs) synthesized by using different diisocyanates and NCO/OH mole ratios, with all WUOs exhibiting a milky-white color.

### 3.4. Basic Properties of WUOs

The basic properties of WUOs synthesized by using two types of diisocyanates and different NCO/OH mole ratios are shown in Table 2. The theoretical solid content of HDI-0.7, HDI-0.8, IPDI-0.7, and IPDI-0.8 was 40%. However, the actual solid contents were in the range of 38.6%–40.0%. It was also observed that the phase inversion points of HDI-0.9 and IPDI-0.9 occurred later compared with other WUOs. At the theoretical solid content of 40%, the viscosities of the respective WUO dispersions of HDI-0.9 and IPDI-0.9 were still high, at 5845 and 4548 cps, which was unfavorable for finishing operations. Therefore, a lower theoretical content of 30% was used, while the actual solid contents were 30.3% and 30.1%, respectively. In addition, the pH values of the WUOs increased as the NCO/OH mole ratio increased. The pH values of HDI-0.7, HDI-0.8, and HDI-0.9 were 8.2, 8.4, and 8.5, respectively, while a similar increasing trend was observed in the pH values of IPDI-0.7, IPDI-0.8, and IPDI-0.9, which were 7.8, 7.9, and 8.0, respectively. All WUOs were weakly alkaline, and the pH range was 7.8–8.5.

The particle size distributions of the WUOs obtained by dynamic light scattering (DLS) are shown in Figure 8 and Figure 9, and the results are summarized in Table 2. For the WUOs synthesized by HDI, the particle size increased with an increase in the NCO/OH mole ratio. HDI-0.7 had the lowest z-average diameter of 427 mm and a PDI of 0.534, followed by HDI-0.8 with a z-average diameter of 581 mm and a PDI of 0.498; HDI-0.9 had the highest z-average diameter of 719 nm and a PDI of 0.534. A similar particle size trend was observed for WUO dispersions synthesized by IPDI. However, the particle sizes were larger, ranging between 441 and 995 mm, while PDI values were within 0.218–0.565. According to a study [26], the rigidity of the molecular chains of WPU increases with hard segment content, which makes the dispersed particles harder to deform under shear force. Therefore, the dispersion particle sizes were larger for the WUOs synthesized by IPDI, which were more rigid than the HDI. As shown in Table 1, the molecular weights of COOH-containing prepolymers increased with an increase in the NCO/OH mole ratio. In the prepolymers with higher molecular weights and rigidity character, the hydrophilic groups had a higher tendency to be packed within molecular chains. Therefore, the formation of uniform, small-sized dispersion particles under shear force during the water dispersion process was more difficult. As a result, the particle size analysis showed that WUOs synthesized by using either HDI or IPDI had the largest particle size with an NCO/OH mole ratio of 0.9. In addition, although the molecular weights of the prepolymers synthesized by using HDI were higher than those synthesized by IPDI, the molecular chains were more flexible and had lower rigidities. Therefore, for the same NCO/OH ratio, the particle sizes of the WUOs synthesized by HDI were smaller than those of the WUOs synthesized by IPDI. For instance, the particle size of HDI-0.9 was 719 nm, which was smaller than the particle size of 995 nm for IPDI-0.9. In the particle size distribution diagrams, signals were observed for particle sizes of 2000–5000 nm, which indicates that the WUOs contained larger particles. In general, the particle size of waterborne coatings is influenced by multiple factors, including ionic group content, molecular weight, and hard segment content. In a study on the synthesis of waterborne coatings using rapeseed oil, IPDI, and DMPA [27], it was reported that the influence of DMPA content on particle size was negligible when the DMPA content in the prepolymer exceeded 4.4 wt %. The DMPA content of the WUO dispersions synthesized in the present study was in the range of 7.3–10.3 wt %. Therefore, the differences in the particle sizes of the WUOs were mainly influenced by the molecular weights and hard segment contents, while the influences of ionic group contents were insignificant. As a result, the particle size distribution for the NCO/OH mole ratio of 0.9 was wider than that of the mole ratio of 0.7, and higher-intensity signals were measured within the range of 2000–5000 nm.

The synthesized WUOs were respectively placed in 1 L polyethylene bottles and observed for signs of gelation, precipitation, discoloration, or stratification. The durations for which the WUOs remained stable are shown in Table 2. The storage stability of WUOs synthesized by HDI increased as the NCO/OH mole ratio increased, with the storage stability of HDI-0.7, HDI-0.8, and HDI-0.9 being 4.5, 4.8, and 5 months, respectively. In contrast, the storage stability of the WUOs synthesized by IPDI decreased as the NCO/OH ratio increased, with the storage stability of IPDI-0.7 and IPDI-0.9 being 2.6 months and 1.5 months, respectively. Therefore, the WUOs synthesized by HDI had higher storage stabilities than those synthesized by IPDI. This can be attributed to the higher flexibilities and molecular weights of the WUO molecules synthesized with HDI. Owing to the hydrophilic groups on the molecular chains are being harder to trap in the dispersion particles, as well as the entanglement, led to greater cohesive forces within the particles, resulting in particles that are hardly to be aggregated by Brownian motion. However, for WUOs synthesized by IPDI, the molecular chains had higher rigidities, and molecular arrangements provided strong steric hindrance. As a result, hydrophilic groups were trapped within the molecular chains, and could not participate in hydration reactions, which resulted in weaker cohesive forces within the dispersion particles. Therefore, the particles could be more readily aggregated by Brownian motion, which led to a higher possibility of stratification.

The rheological properties of the WUOs were measured by a rheometer, and the results are shown in Figure 10 and Figure 11. Rheological properties refer to the deformation and flow characteristics that arise when a material is subjected to an external force, and are usually determined by a rheometer to measure the variations in viscosity, i.e., the fluid’s resistance to deformation. As the viscosities of the WUOs synthesized in the present study decreased with shear rate increase, they possessed shear-thinning behaviors characteristic of pseudoplastic fluids. Therefore, they can effectively prevent the coating from dripping during finishing.

## 4. Conclusions

In the present study, transesterification of LO with GL was performed to obtain OH-containing LOG, which were subsequently reacted using different NCO/OH mole ratios (0.7, 0.8, and 0.9) with acetone and DMPA, HDI, or IPDI, to obtain COOH-containing prepolymers. The prepolymers were then neutralized with TEA to form ionomers, and were dispersed into water to form water–acetone dispersions. Finally, acetone removal was performed by vacuum distillation to obtain WUOs. From the FTIR spectra of the prepolymers synthesized by HDI and IPDI, it was found that the intensity of the OH absorption peak decreased, the NCO absorption peak was absent, and characteristic absorption peaks corresponding to the urethane structure and COOH were presented, which show that the diisocyanates, LOG, and DMPA successfully reacted to form prepolymers. In addition, the Mw and polydispersity of the prepolymers increased with an increase in the NCO/OH mole ratio. After neutralization of the ionomer solution with TEA, the phase inversion point during the water dispersion process occurred later when the NCO/OH mole ratio was increased, which led to a decrease in solid content. The WUOs obtained after acetone removal had milky-white appearances, were weakly alkaline, and exhibited pseudoplastic behaviors. For the WUOs synthesized by HDI, particle size and storage stability increased as the NCO/OH mole ratio increase. Similarly, the particle sizes of the WUOs synthesized by IPDI increased with increasing NCO/OH mole ratio. However, the storage stability decreased as the NCO/OH mole ratio increased. In addition, the films properties of the WUOs for wood coatings have been examined and will be submitted in the future report.

## Figures and Tables

**Figure 1 polymers-10-01235-f001:**
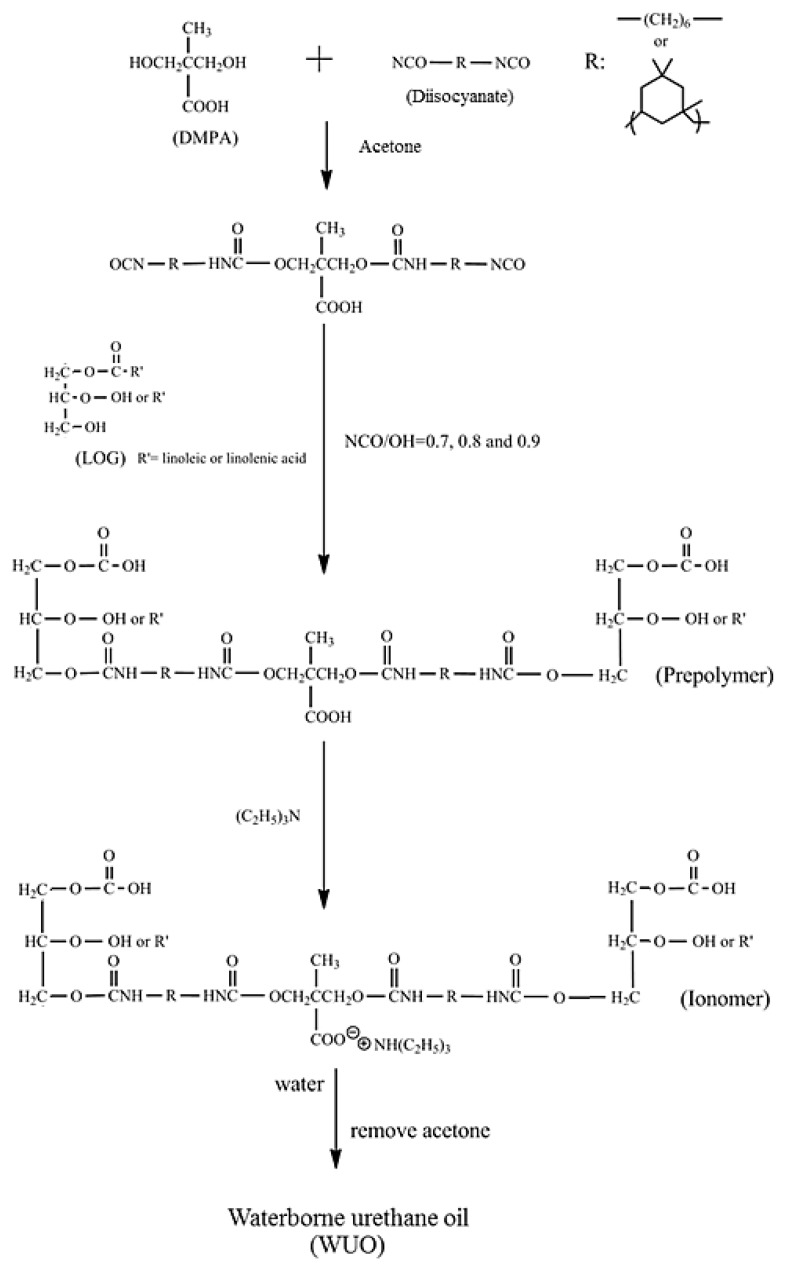
Synthesis process of waterborne urethane oil (WUO).

**Figure 2 polymers-10-01235-f002:**
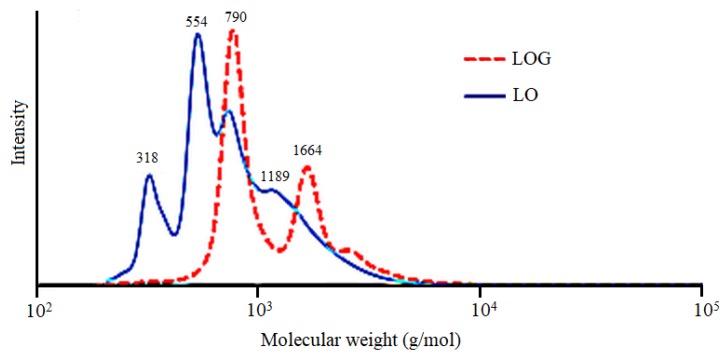
Gel permeation chromatography (GPC) diagrams of LO and LOG.

**Figure 3 polymers-10-01235-f003:**
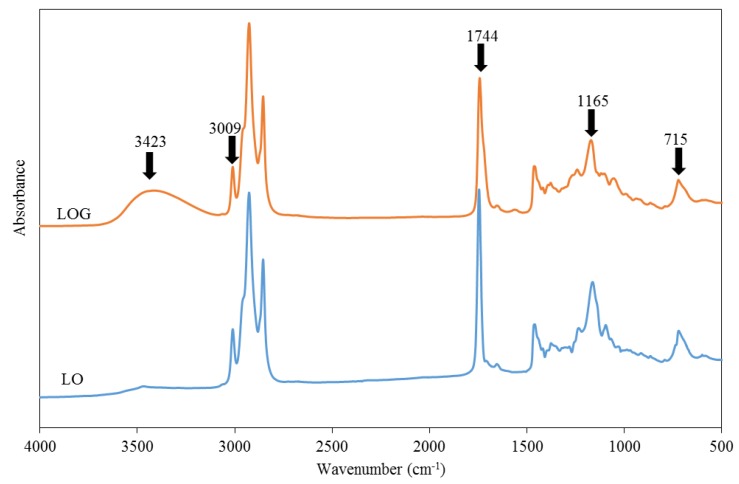
FTIR spectra of linseed oil (LO) and linseed oil glyceride (LOG).

**Figure 4 polymers-10-01235-f004:**
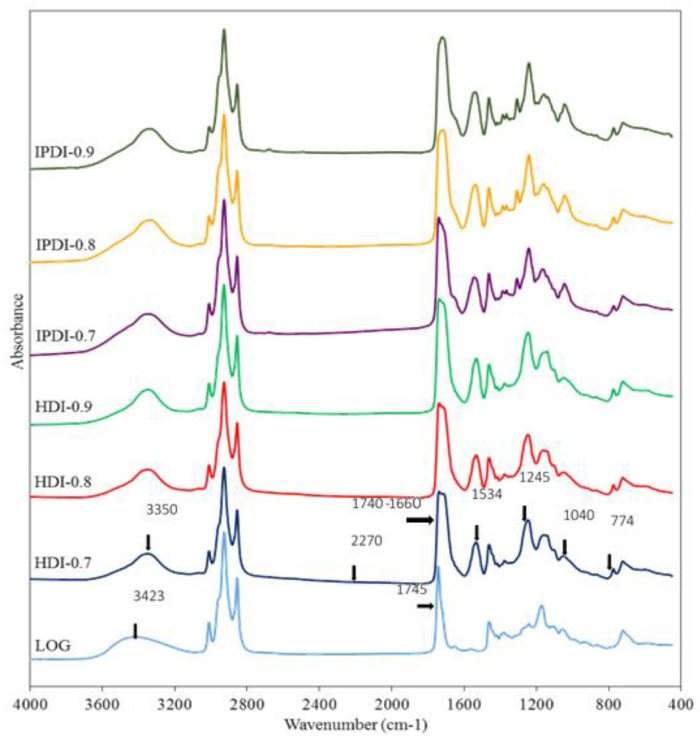
FTIR spectra of LOG and COOH-containing prepolymers synthesized using different NCO/OH mole ratios.

**Figure 5 polymers-10-01235-f005:**
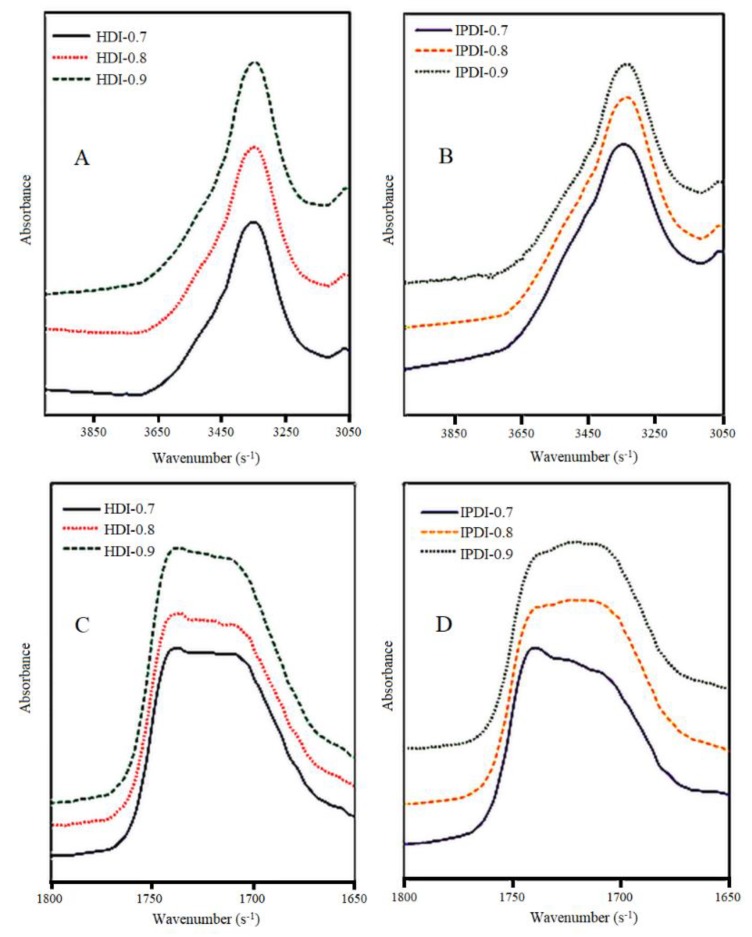
FTIR spectra of COOH-containing prepolymers synthesized using different isocyanates and NCO/OH mole ratios in the 4000–3050 cm^−1^ range for (**A**) hexamethylene diisocyanate (HDI) and (**B**) isophorone diisocyanate (IPDI), and in the 1800–1650 cm^−1^ range for (**C**) HDI and (**D**) IPDI.

**Figure 6 polymers-10-01235-f006:**
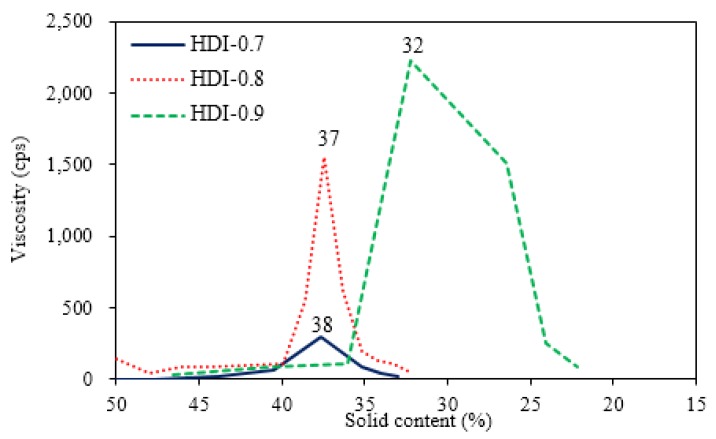
Viscosity variation of HDI ionomer dispersions with different NCO/OH mole ratios during the water dispersion process.

**Figure 7 polymers-10-01235-f007:**
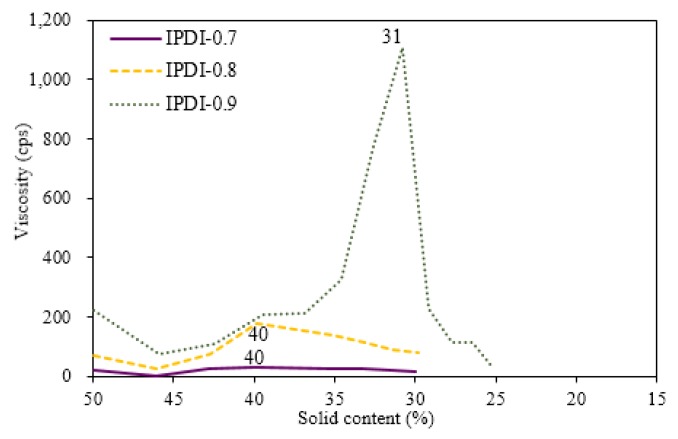
Viscosity variation of IPDI ionomer dispersions with different NCO/OH mole ratios during water dispersion process.

**Figure 8 polymers-10-01235-f008:**
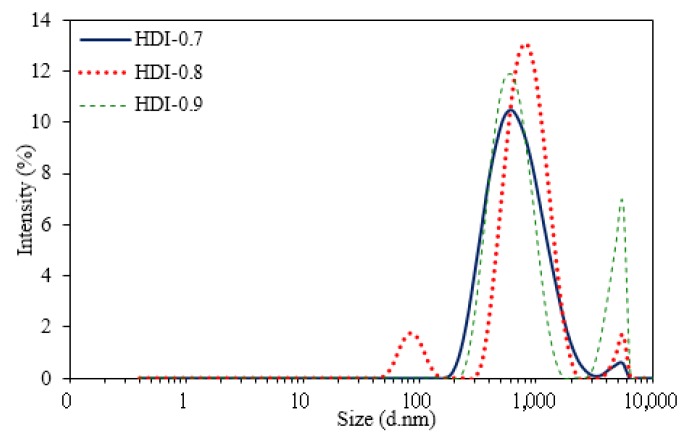
Particle size distributions of WUOs synthesized using HDI and different NCO/OH mole ratios.

**Figure 9 polymers-10-01235-f009:**
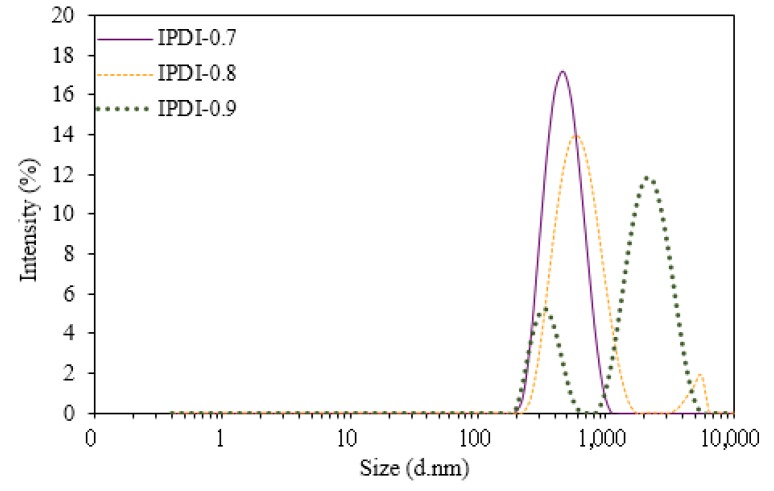
Particle size distributions of WUOs synthesized using IPDI and different NCO/OH mole ratios.

**Figure 10 polymers-10-01235-f010:**
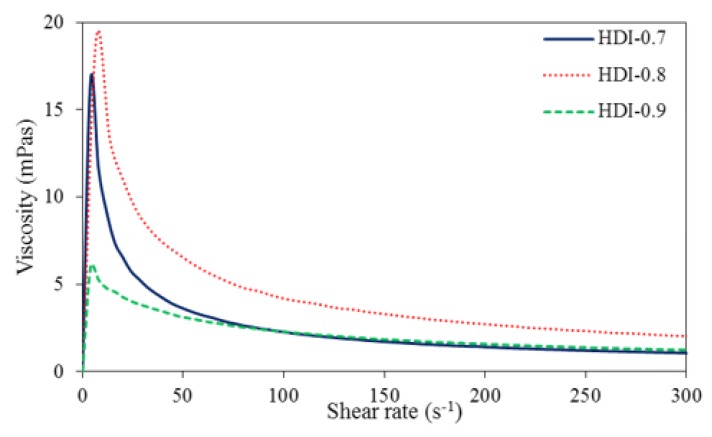
Rheological behaviors of WUOs synthesized using HDI and different NCO/OH mole ratios.

**Figure 11 polymers-10-01235-f011:**
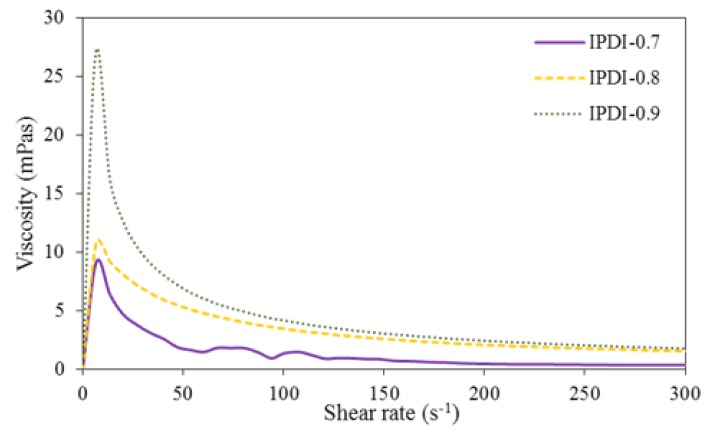
Rheological behaviors of WUOs synthesized using IPDI and different NCO/OH mole ratios.

**Table 1 polymers-10-01235-t001:** Molecular weight parameters of COOH-containing prepolymers synthesized using different isocyanates and NCO/OH mole ratios.

Prepolymer	*M*_w_ (g/mol)	*M*_n_ (g/mol)	Polydispersity (*M*_w_/*M*_n_)
HDI-0.7	4214	1631	2.6
HDI-0.8	7181	1678	4.3
HDI-0.9	10,734	1682	6.4
IPDI-0.7	2962	1496	2.0
IPDI-0.8	5861	1776	3.3
IPDI-0.9	6126	1418	4.3

**Table 2 polymers-10-01235-t002:** Basic properties of WUOs synthesized using different isocyanates and NCO/OH mole ratios.

WUO	Solid Content (%)	pH	Z-Average ^a^ (nm)	PDI ^b^	Stability (Months)
HDI-0.7	38.6 ± 0.2	8.2	427	0.534	4.5
HDI-0.8	40.0 ± 0.3	8.4	581	0.498	4.8
HDI-0.9	30.3 ± 0.1	8.5	719	0.534	5.0
IPDI-0.7	38.8 ± 0.4	7.8	441	0.218	2.6
IPDI-0.8	39.6 ± 0.1	7.9	586	0.329	2.0
IPDI-0.9	30.1 ± 0.1	8.0	995	0.565	1.5

^a^ Cumulative mean. ^b^ Polydispersity index.

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
