# Peer review of "Synthesis of Linseed Oil-Based Waterborne Urethane Oil Wood Coatings"

_polymers, 2018, doi:10.3390/polym10111235_

Round 1

Reviewer 1 Report

The manuscript authored by Lu et al reports a series of linseed oil waterborne polyurethanes for wood coatings. The synthesis process is a traditional acetone method with modulation of kinds of diisocyanate and the ratios of –NCO and –OH. The characterizations have been fully conducted. However, there are some revisions should be amended.

1.     The symbol of temperature in the whole manuscript should be corrected.

2.     How did author purify the monoglyceride to rule out diglyceride and triglyceride? There is no description in the part of "Preparation of linseed oil glyceride". If LOG is used without purification, how do the authors guarantee the experimental repeatability?

3.     The structure labeled by red circle do not line up with bonding times of carbon atoms. The corrected structure should be offered.

4.     The description of the sentence of " However, all these fatty acids have not contained 149 hydroxyl (OH) groups, and cannot react with diisocyanate." is not strictly accurate. Actually, the reaction between carboxyl and isocyanate groups is allowed. The reason why the carboxyls in DMPA do not react with isocyante is steric effect of two hydroxyl groups.

5.     There are no any references cited in the part of Introduction. The authors should insert proper literatures in this part. In addition, as for waterborne polyurethane studies, the following literatures should be quoted.

Materials, 2017, 10, 1247; Polymer Chemistry, 2018, 9, 1303-1308

Author Response

Response to Referee’s Comments and Suggestions

Response to Referee

We thank the reviewer for carefully reading our manuscript and for providing constructive comments. In the revised version, we have adapted our article following your comments and suggestions.

1. The symbol of temperature in the whole manuscript should be corrected.

Ans: We have corrected these mistakes in whole manuscript.

2. How did author purify the monoglyceride to rule out diglyceride and triglyceride? There is no description in the part of "Preparation of linseed oil glyceride". If LOG is used without purification, how do the authors guarantee the experimental repeatability?

Ans: Thank you for your helpful advice. The LOG was not further purified in this study. In our previous study, we identified the LOG components by gel permeation chromatography, and we found there was no unreactive linseed oil/ triglyceride in the LOG. We had sign the following study in this part. “15. Chang, C.W., K.T. Lu. Organic-inorganic hybrid linseed oil-based urethane oil wood coatings. 2017, 134, 44562. DOI: 10.1002/app.44562. https://onlinelibrary.wiley.com/doi/abs/10.1002/app.44562)”

3. The structure labeled by red circle do not line up with bonding times of carbon atoms. The corrected structure should be offered.

Ans: We have corrected this mistake of Fig.1 The Fig. 1 can be found in uploaded file. 

4. The description of the sentence of " However, all these fatty acids have not contained hydroxyl (OH) groups, and cannot react with diisocyanate." is not strictly accurate. Actually, the reaction between carboxyl and isocyanate groups is allowed. The reason why the carboxyls in DMPA do not react with isocyante is steric effect of two hydroxyl groups.

Ans: Thank you for your considerate advice. The linseed oil, as a triglyceride, was formed by esterification of the different fatty acids and glycerol. The “fatty acids” were used to describe the different 18C-alkyl components in the triglyceride structure in this study. We have improved this sentence as following.

Original:

“However, all these fatty acids have not contained hydroxyl (OH) groups, and cannot react with diisocyanate.”

Corrected:

“However, the linseed oil does not contain hydroxyl (OH) groups, and cannot react with diisocyanate.”

5. There are no any references cited in the part of Introduction. The authors should insert proper literatures in this part. In addition, as for waterborne polyurethane studies, the following literatures should be quoted.

Ans: We are appreciated for your suggestion. We have added many references in        the related sentences. The added references are listed below.

1. Islam, M.R.; Beg, M.D.H.; Jamari, S.S. Development of Vegetable‐Oil‐Based Polymers. J. Appl. Polym. Sci. 2014, 131, 40787-40799, DOI: 10.1002/app.40787, Available online: https://onlinelibrary.wiley.com/doi/full/10.1002/app.40787 (accessed on 16 April 2014).

2. Pergal, M.V.; Džunuzović, J.V.; Poręba, R.; Ostojić, S.; Radulović, A.; Špírková, M. Preparation and Characterization of Polyurethanes with Cross-Linked Siloxane in the Side Chain by Sol-Gel Reactions. Prog. Org. Coat. 2013, 76, 743-756. DOI: 10.3390/ma10030247, Available online: https://www.mdpi.com/1996-1944/10/3/247 (accessed on 2017 Feb 28).

3. Zhang, C.; Xia, Y.; Chen, R.; Huh, S.; Johnston, P.A.; Kessler, M.R. Soy-Castor Oil Based Polyols Prepared Using a Solvent-free and Catalyst-free Method and Polyurethanes. Therefrom. Green Chem. 2013, 15, 1477-1484. DOI: 10.1039/C3GC40531A, Available online: https://pubs.rsc.org/en/content/articlelanding/2013/gc/c3gc40531a#!divAbstract (accessed on 12 Apr 2013).

4. Ghosh, B.; Gogoi, S.; Thakur, S.; Karak, N. Bio-based Waterborne Polyurethane/Carbon Dot Nanocomposite as a Surface Coating Material. Prog. Org. Coat. 2016, 90, 324-330. DOI: j.porgcoat.2015.10.025, Available online: https://www.sciencedirect.com/science/article/pii/S0300944015302587?via%3Dihub (accessed on 21 November 2015).

5. Wang, T.; Sun, W.; Zhang, X.; Xu, H.; Xu, F. Waterborne Polyurethane Coatings with Covalently Linked Black Dye Sudan Black B. Materials 2017, 10, 1247-1260, DOI: 10.3390/ma10111247, Available online: https://www.mdpi.com/1996-1944/10/11/1247 (accessed on 28 Oct 2017).

6. Wang, T.; Zhou, C.; Zhang, X.; Xu, D. Waterborne Polyurethanes Prepared from Benzophenone Derivatives with Delayed Fluorescence and Room-Temperature Phosphorescence. Polym. Chem. 2018, 9, 1303-1308, DOI: 10.1039/C7PY01995E, Available online: https://pubs.rsc.org/en/content/articlelanding/2018/py/c7py01995e#!divAbstract (accessed on 12 Feb 2018).

7. Xia, Y.; Larock, R.C. Vegetable Oil-based Polymeric Materials: Synthesis, Properties, and Applications. Green Chem. 2010, 12, 1893-1909. DOI: 10.1039/C0GC00264J, Available online: https://pubs.rsc.org/en/Content/ArticleLanding/2010/GC/c0gc00264j#!divAbstract (accessed on 01 Oct 2010).

8. Chen, Y.C.; Tai, W. Castor Oil-Based Polyurethane Resin for Low-Density Composites with Bamboo Charcoal. Polymers, 2018, 10, 1100-1112, DOI:10.3390/polym10101100, Available online: https://www.mdpi.com/2073-4360/10/10/1100 (accessed on 5 October 2018).

9. Pfister, D.P.; Xia, Y.; Larock, R.C. Recent Advances in Vegetable Oil-Based Polyurethanes. Chem. Sus. Chem. 2011, 4, 703-717, DOI: 10.1002/cssc.201000378, Available online: https://onlinelibrary.wiley.com/doi/abs/10.1002/cssc.201000378 (accessed on 20 May 2011).

10. Veigel, S.; Lems, E.M.; Grüll, G.; Hansmann, C.; Rosenau, T.; Zimmermann, T.; Gindl-Altmutter, W. Simple Green Route to Performance Improvement of Fully Bio-Based Linseed Oil Coating Using Nanofibrillated Cellulose. Polymers, 2017, 9, 425-438, DOI: 10.3390/polym9090425, Available online: https://www.mdpi.com/2073-4360/9/9/425 (accessed on 7 Sep 2017).

11. Li, S.; Xu, C.; Yang, W.; Tang, Q. Thermoplastic Polyurethanes Stemming from Castor Oil: Green Synthesis and Their Application in Wood Bonding. Coatings, 2017, 7, 159-169, DOI: 10.3390/coatings7100159, Available online: https://www.mdpi.com/2079-6412/7/10/159 (accessed on 12 Sep 2017).

12. Ahn, B.K.; Kraft, S.; Wang, D.; Sun, X.S. Thermally Stable, Transparent, Pressure-Sensitive Adhesives from Epoxidized and Dihydroxyl Soybean Oil. Biomacromolecules, 2011, 12, 1839-1843, DOI: 10.1021/bm200188u, Available online: https://pubs.acs.org/doi/abs/10.1021/bm200188u (accessed on 17 March 2011).

Reviewer 2 Report

First of all, English language needs to be checked, bunch of grammar mistakes should be corrected. 

Introduction part must be definitely improved. This part of manuscript has to describe state of art in particular area, but Authors did not include any literature reference and it is hard for me to think that there are no papers published related to the urethane-based wood coatings. 

Symbol of Celsius degrees has to be corrected. 

The measurement of hydroxyl number should be described in manuscript. Same for solid content. 

Sentence "In addition, a signal at 1,664 g/mol of the LOG curve corresponding to the linseed oil dimer which was not found in the LO curve." makes no sense.

How was theoretical value of LOG calculated?

It would be useful for analysis if Authors would present FTIR spectra for isocyanates and DMPA in the Fig. 4. 

When describing the GPC results, Authors should give some explanation related to the differences in Mw and Mn between HDI- and IPDI-based prepolymers. 

Authors definitely have to use more literature references when describing the results. Need to compare their results with the results obtained by other researchers. There is quite a lot works related to waterborne polyurethane coatings. 

Author Response

Response to Referee’s Comments and Suggestions

Response to Referee

We thank the reviewer for carefully reading our manuscript and for providing constructive comments. In the revised version, we have adapted our article following your comments and suggestions.

1. First of all, English language needs to be checked, bunch of grammar mistakes should be corrected.

Ans: The referee’s suggestion is noted. We have requested a native speaker to review our paper again, and it should be well organized in the revised version.

2. Introduction part must be definitely improved. This part of manuscript has to describe state of art in particular area, but Authors did not include any literature reference and it is hard for me to think that there are no papers published related to the urethane-based wood coatings.

Ans: Thank you for your considerate advice. We have improved the introduction, and about 12 reference papers were added to the related text.

3. Symbol of Celsius degrees has to be corrected.

Ans: We have corrected these mistakes in whole manuscript.

4. The measurement of hydroxyl number should be described in manuscript. Same for solid content. hydroxyl

Ans: The referee’s suggestion is noted. We had improved the description of measurements as following.

“The hydroxyl numbers were determined by using the acetic anhydride/pyridine method in accordance with the Standard Test Method for Hydroxyl Value ofFatty Oils and Acids (ASTM D1957).”

“Solid contents were determined in accordance with CNS 5133 (Chinese National Standards, Taiwan). The mass retention of 3 g sample in the 105oC oven for 3 h was determined.”

5. Sentence "In addition, a signal at 1,664 g/mol of the LOG curve corresponding to the linseed oil dimer which was not found in the LO curve." makes no sense.

Ans: We have adapted our paper following your comment and removed this sentence.

6. How was theoretical value of LOG calculated?

Ans: Both of following ways were used to calculate the theoretical hydroxyl value of LOG. (1) We calculated the LOG theoretical value by the molecular weights of linseed oil and glycerol with a mole ratio of 1:1. (2) We determined the hydroxyl values of linseed oil and glycerol, respectively, and calculated the LOG theoretical hydroxyl value by experimental values with a mole ratio of 1:1. However, we found that the theoretical hydroxyl values from the two ways are very similar. At last, we used the (1) method to calculate the theoretical value of LOG.

7. It would be useful for analysis if Authors would present FTIR spectra for isocyanates and DMPA in the Fig. 4.

Ans: We are appreciated for your suggestion. The FTIR characteristic peaks of isocyanates are NCO, alkyl groups, and characteristic peaks of DMPA are hydroxyl groups, carboxylic acid, alkyl groups. Considering to the length of Fig. 4, we had described these reactive characteristic peaks in the text.

8. When describing the GPC results, Authors should give some explanation related to the differences in Mw and Mn between HDI- and IPDI-based prepolymers.

Ans: We have added a paragraph to explain the differences between HDI- and IPDI-based prepolymers as following.

Due to the reactive difference of two NCO groups between HDI and IPDI, the IPDI-based prepolymer which was obtained through two-step addition reaction is more homogeneous than the HDI-based prepolymer. Therefore, the IPDI-based prepolymers had a lower Mw and polydispersity than the HDI-based prepolymers.”

9. Authors definitely have to use more literature references when describing the results. Need to compare their results with the results obtained by other researchers. There is quite a lot works related to waterborne polyurethane coatings.

Ans: In this study, we focused on the relations of synthesized process with different raw material composition. We had discussed the experimental phenomena with many related references. In the next paper, we have further compared the coating and film properties with other literatures. We are appreciated to follow suggestions to add more literature references about waterborne polyurethane coatings in the introduction, and these papers will provide readers with more additional information to assist our study.

Reviewer 3 Report

This manuscript presents the synthesis and characterization of urethane-containing coating incorporating linseed oil. In general, the manuscript is well-written. The characterization aspect of the resulting materials is thorough and well explained. 

It would be useful if the authors discuss in their introduction if other researchers have performed similar works to distinguish how their study is any different than the rest and highlighting the motivation of the present work. I'm quite surprised there is no citation in the introduction section, not that it is absolutely necessary but I think this is the section where you'd need the most citations to distinguish and differentiate your work. 

Author Response

Response to Referee’s Comments and Suggestions

Response to Referee

We thank the reviewer for reading our manuscript and for providing constructive comments. In the revised version, we have adapted our article following your comments and suggestions.

1.  This manuscript presents the synthesis and characterization of urethane-containing coating incorporating linseed oil. In general, the manuscript is well-written. The characterization aspect of the resulting materials is thorough and well explained. It would be useful if the authors discuss in their introduction if other researchers have performed similar works to distinguish how their study is any different than the rest and highlighting the motivation of the present work. I'm quite surprised there is no citation in the introduction section, not that it is absolutely necessary but I think this is the section where you'd need the most citations to distinguish and differentiate your work.

Ans: We are appreciated for your suggestion. We have improved the introduction and added many related references in this paper. The added references were listed below.

1. Islam, M.R.; Beg, M.D.H.; Jamari, S.S. Development of Vegetable‐Oil‐Based Polymers. J. Appl. Polym. Sci. 2014, 131, 40787-40799, DOI: 10.1002/app.40787, Available online: https://onlinelibrary.wiley.com/doi/full/10.1002/app.40787 (accessed on 16 April 2014).

2. Pergal, M.V.; Džunuzović, J.V.; Poręba, R.; Ostojić, S.; Radulović, A.; Špírková, M. Preparation and Characterization of Polyurethanes with Cross-Linked Siloxane in the Side Chain by Sol-Gel Reactions. Prog. Org. Coat. 2013, 76, 743-756. DOI: 10.3390/ma10030247, Available online: https://www.mdpi.com/1996-1944/10/3/247 (accessed on 2017 Feb 28).

3. Zhang, C.; Xia, Y.; Chen, R.; Huh, S.; Johnston, P.A.; Kessler, M.R. Soy-Castor Oil Based Polyols Prepared Using a Solvent-free and Catalyst-free Method and Polyurethanes. Therefrom. Green Chem. 2013, 15, 1477-1484. DOI: 10.1039/C3GC40531A, Available online: https://pubs.rsc.org/en/content/articlelanding/2013/gc/c3gc40531a#!divAbstract (accessed on 12 Apr 2013).

4. Ghosh, B.; Gogoi, S.; Thakur, S.; Karak, N. Bio-based Waterborne Polyurethane/Carbon Dot Nanocomposite as a Surface Coating Material. Prog. Org. Coat. 2016, 90, 324-330. DOI: j.porgcoat.2015.10.025, Available online: https://www.sciencedirect.com/science/article/pii/S0300944015302587?via%3Dihub (accessed on 21 November 2015).

5. Wang, T.; Sun, W.; Zhang, X.; Xu, H.; Xu, F. Waterborne Polyurethane Coatings with Covalently Linked Black Dye Sudan Black B. Materials 2017, 10, 1247-1260, DOI: 10.3390/ma10111247, Available online: https://www.mdpi.com/1996-1944/10/11/1247 (accessed on 28 Oct 2017).

6. Wang, T.; Zhou, C.; Zhang, X.; Xu, D. Waterborne Polyurethanes Prepared from Benzophenone Derivatives with Delayed Fluorescence and Room-Temperature Phosphorescence. Polym. Chem. 2018, 9, 1303-1308, DOI: 10.1039/C7PY01995E, Available online: https://pubs.rsc.org/en/content/articlelanding/2018/py/c7py01995e#!divAbstract (accessed on 12 Feb 2018).

7. Xia, Y.; Larock, R.C. Vegetable Oil-based Polymeric Materials: Synthesis, Properties, and Applications. Green Chem. 2010, 12, 1893-1909. DOI: 10.1039/C0GC00264J, Available online: https://pubs.rsc.org/en/Content/ArticleLanding/2010/GC/c0gc00264j#!divAbstract (accessed on 01 Oct 2010).

8. Chen, Y.C.; Tai, W. Castor Oil-Based Polyurethane Resin for Low-Density Composites with Bamboo Charcoal. Polymers, 2018, 10, 1100-1112, DOI:10.3390/polym10101100, Available online: https://www.mdpi.com/2073-4360/10/10/1100 (accessed on 5 October 2018).

9. Pfister, D.P.; Xia, Y.; Larock, R.C. Recent Advances in Vegetable Oil-Based Polyurethanes. Chem. Sus. Chem. 2011, 4, 703-717, DOI: 10.1002/cssc.201000378, Available online: https://onlinelibrary.wiley.com/doi/abs/10.1002/cssc.201000378 (accessed on 20 May 2011).

10. Veigel, S.; Lems, E.M.; Grüll, G.; Hansmann, C.; Rosenau, T.; Zimmermann, T.; Gindl-Altmutter, W. Simple Green Route to Performance Improvement of Fully Bio-Based Linseed Oil Coating Using Nanofibrillated Cellulose. Polymers, 2017, 9, 425-438, DOI: 10.3390/polym9090425, Available online: https://www.mdpi.com/2073-4360/9/9/425 (accessed on 7 Sep 2017).

11. Li, S.; Xu, C.; Yang, W.; Tang, Q. Thermoplastic Polyurethanes Stemming from Castor Oil: Green Synthesis and Their Application in Wood Bonding. Coatings, 2017, 7, 159-169, DOI: 10.3390/coatings7100159, Available online: https://www.mdpi.com/2079-6412/7/10/159 (accessed on 12 Sep 2017).

12. Ahn, B.K.; Kraft, S.; Wang, D.; Sun, X.S. Thermally Stable, Transparent, Pressure-Sensitive Adhesives from Epoxidized and Dihydroxyl Soybean Oil. Biomacromolecules, 2011, 12, 1839-1843, DOI: 10.1021/bm200188u, Available online: https://pubs.acs.org/doi/abs/10.1021/bm200188u (accessed on 17 March 2011).

Round 2

Reviewer 2 Report

I'm ok with the changes made by Authors.